# Repurposing Based Identification of Novel Inhibitors against MmpS5-MmpL5 Efflux Pump of *Mycobacterium smegmatis*: A Combined In Silico and In Vitro Study

**DOI:** 10.3390/biomedicines10020333

**Published:** 2022-01-31

**Authors:** Mohd Shahbaaz, Dmitry A. Maslov, Aleksey A. Vatlin, Valery N. Danilenko, Maria Grishina, Alan Christoffels

**Affiliations:** 1South African Medical Research Council Bioinformatics Unit, South African National Bioinformatics Institute, University of the Western Cape, Private Bag X17, Bellville 7535, Cape Town, South Africa; mohammed.shahbaaz@gmail.com; 2Laboratory of Bacterial Genetics, Vavilov Institute of General Genetics Russian Academy of Sciences, 119333 Moscow, Russia; maslov_da@vigg.ru (D.A.M.); vatlin_alexey123@mail.ru (A.A.V.); valerid@vigg.ru (V.N.D.); 3Institute of Ecology, Peoples’ Friendship University of Russia (RUDN University), 117198 Moscow, Russia; 4Laboratory of Computational Modeling of Drugs, Higher Medical and Biological School, South Ural State University, P. Lenina 76, 454080 Chelyabinsk, Russia; grishinama@susu.ru; 5Africa Centres for Disease Control and Prevention, African Union Headquarters, Addis Ababa W21K19, Ethiopia

**Keywords:** tuberculosis, MmpS5-MmpL5 efflux pumps, drug resistance, virtual screening, QSAR, molecular dynamics simulations

## Abstract

In the current era of a pandemic, infections of COVID-19 and Tuberculosis (TB) enhance the detrimental effects of both diseases in suffering individuals. The resistance mechanisms evolving in *Mycobacterium tuberculosis* are limiting the efficiency of current therapeutic measures and pressurizing the stressed medical infrastructures. The bacterial efflux pumps enable the development of resistance against recently approved drugs such as bedaquiline and clofazimine. Consequently, the MmpS5-MmpL5 protein system was selected because of its role in efflux pumping of anti-TB drugs. The MmpS5-MmpL5 systems of *Mycobacterium smegmatis* were modelled and the virtual screening was performed using an ASINEX library of 5968 anti-bacterial compounds. The inhibitors with the highest binding affinities and QSAR based highest predicted inhibitory concentration were selected. The MmpS5-MmpL5 associated systems with BDE_26593610 and BDD_27860195 showed highest inhibitory parameters. These were subjected to 100 ns Molecular Dynamics simulations and provided the validation regarding the interaction studies. The in vitro studies demonstrated that the BDE_26593610 and BDD_27860195 can be considered as active inhibitors for *M. smegmatis* MmpS5-MmpL5. The outcomes of this study can be utilized in other experimentation aimed at drug design and discovery against the drug resistance strains of *M. tuberculosis*.

## 1. Introduction

The increasing evolution of drug resistance during tuberculosis (TB) infection has made it a global challenge [1,2]. The multidrug and extensive drug-resistance strains (MDR- and XDR-TB) of *Mycobacterium tuberculosis* resulted in reduced efficiency of available treatment [3], which makes it necessary to formulate new potent drug molecules [4,5]. Additionally, in combination with TB, the illnesses associated with the COVID-19 pandemic are usually manifested in the respiratory systems and the severity result in system failures [6]. In combination with the COVID-19 global occurrence, the other epidemics are propagating in a parallel manner and it is imperative not to forget the conditions resulting from the infection of *M. tuberculosis* [6]. The studies suggest that the patients with COVID-19 are getting infected with *M. tuberculosis* in a synergistic manner which resulted in the enhancement of detrimental effects of both diseases [6]. This may limit the treatment of both the diseases, in particular TB, which has a history of drug resistance resulting from the activities of a variety of mechanisms.

The role of efflux pumps in the development of drug resistance in *M. tuberculosis* is one of the arising problems, especially due to their involvement in bedaquiline tolerance, which is the first anti-TB drug approved by FDA in 40 years [7]. Bedaquiline is found to be very effective in treating the drug resistant strains of *M. tuberculosis* and its newer regimens reduced the period of treatment and increased its effectiveness. Yet, there is an alarming emergence of strains with resistance to the anti-TB drugs such as clofazimine along with bedaquiline, mediated by the efflux pump mechanisms [7]. The recent proceedings showed that the use of efflux pump inhibitors like verapamil decreases the inhibitory concentration of the administered drugs as well as enables the reduction of their toxicological effects and the frequency of the cross resistance [8]. The inhibitors targeting the efflux pumps impose a range of influences on the phenotypes associated with the drug susceptibility in a variety of studied *M. tuberculosis* strains [8]. As reported in drug-susceptible strains of bacteria, inhibitors such as reserpine and 2,4-dinitrophenol have diminishing effects on the accumulation of drugs like rifampicin, norfloxacin, while the verapamil in combination with bedaquiline reduces its minimum inhibitory concentration (MIC) [8]. The efficiency of the efflux inhibitors depends on a variety of experimental factors including their chemical nature and genetics of the *M. tuberculosis* strain [9]. 

It is evident from various studies that the effects of the efflux pumps are wider than limiting to a single drug. For example, after *M. tuberculosis* infection of macrophages tolerance was developed for a variety of anti-TB drugs such as isoniazid, rifampicin, bedaquiline, and moxifloxacin [9]. A hypothesis was formulated that highlighted the role of efflux pumps in the acquisition of drug resistance conferring mutations through the development of low concentration mediated tolerance from the activities of the pumps [10]. These observations highlighted the role of efflux pumping in the development of drug resistance and therefore can be considered a promising drug target in the development of efficient anti-TB therapies. 

The mycobacterial MmpS5-MmpL5 efflux system was found to have a role in the development of tolerance against anti-TB drugs and drug candidates [11] in different mycobacterial species, including bedaquiline, clofazimine, and azoles in *M. tuberculosis* [7,12], thiacetazone derivatives in *M. abscessus* [13], and imidazo[1,2-b][1,2,4,5]tetrazines, and tryptanthrins in *M. smegmatis* [14,15]. In *M. tuberculosis* H37Rv strain, there are 14 genes expressed to form MmpL proteins which showed structural similarities to the members of HAE family of efflux proteins which belong to the RND superfamily of transporter proteins [11]. Consequently, the proteins of MmpS5-MmpL5 system of *M. smegmatis* were selected and showed very high similarity to the respective proteins of *M. tuberculosis*. Subsequently, their three-dimensional (3-D) structures were predicted and subjected to virtual screening with a library of anti-bacterial inhibitors. The inhibitors with the highest binding affinities with the docked MmpS5-MmpL5 system were further analysed using the principles of Molecular Dynamics (MD) simulations. Afterward, the inhibitors were subjected to a series of in vitro experimentations which provided validation to the computational analyses. The outcomes of this study can facilitate novel experimental procedures involved in designing novel anti-TB drugs aimed at inhibiting the efflux pump mechanisms.

## 2. Materials and Methods

### 2.1. Modeling of MmpS5-MmpL5 Heterodimer

The previously published methodology was used for the assembly of MmpS5-MmpL5 system [16]. Due to lower similarity of the MmpS5-MmpL5 protein sequences with the structural homologs, the Phyre2 server was used for the modeling which utilized the threading-based algorithm for the identification of the remote structural homolog and to predict the structure with the higher confidence [17]. The obtained structures were refined using GROMACS 2018-2 molecular mechanics package [18], which was used to energy minimize the systems and for the removal of the steric clashes. The stereo-chemical validity for the predicted models was obtained using the Ramachandran plot. After establishing the quality of the predicted models, the homo-trimer of MmpL5 system was predicted using the SymmDock server [19]. The MmpS5 protein was complexed with energy minimized trimeric MmpL5 protein using the ZDOCK [20] and the resulted heterodimer of MmpS5-MmpL5 system was used for further analyses.

### 2.2. Virtual Screening and Quantitative Structure-Activity Relationship (QSAR)

The primary step for the virtual screening was to proceed with the identification of substrate binding sites using the COACH-D server [21], which is an improved version that utilized the principles of molecular docking to refine the predicted poses. Thereafter, a set containing 5968 anti-bacterial compounds was obtained from the ASINEX screening libraries [22]. The 3-D coordinates of all the compounds were prepared and optimized using “LigPrep” modules of Schrodinger 2020-2 suite [23] and consequently docked into the predicted site of MmpL5 using AutoDock Vina package [24]. On the basis of the outcomes generated using the COACH-D server, the grid spacing of 46 × 48 × 40 was used in XYZ directions. The default values of the parameters were fixed which uses Vina scoring function instead of AD4. The docked inhibitors with the highest free energy of binding were selected and further filtered using QSAR models, which were constructed based on training and test sets which were collected from the literature [25]. The information regarding the structures and inhibitory concentration were collected [25] and the regression modeling was performed using AutoQSAR module, which utilized diverse machine learning algorithms for the modeling of structure-activity relationship [26]. For our study, Kernel partial least squares (KPLS) methods showed most promising results. The KPLS method is among the popular techniques for regression of complex non-linear data sets, with the modeling is performed by mapping the data in a higher dimensional feature space through the kernel transformation. The disadvantage of such a transformation is, however, that information about the contribution of the original variables in the regression is lost.

### 2.3. Molecular Dynamics (MD) Simulations

The docked complexes of compounds with best inhibitory parameters were selected for further structure-based analyses using the GROMACS 2018-2 molecular mechanics package [18]. In the primary steps, the membrane orientation of the MmpS5-MmpL5 and docked inhibitors was predicted using the PPM server [27]. The modeled system was embedded in membrane of 1-palmitoyl-2-oleoyl-sn-glycero-3-phosphocholine (POPC) composition around the oriented sites using the CHARMM-GUI [28]. The generated systems were solvated using the TIP3P water model [29]. The negative charges in the systems were neutralized by the addition of counter K^+^ ions. The CHARMM36 force-field was used for the topology generation of protein and docked inhibitor components [28]. After assembled systems were subjected to the energy minimization using steepest descent algorithm. The CHARMM-GUI based input generator divided the equilibration into six steps, each was carried out for 2 ns time scale. The temperature of 300 K was maintained for the system using Berendsen weak coupling method [30] and in the production stage by Nose-Hoover algorithm [31]. While for all stages the Parrinello-Rahman barostat [32] was used for the maintenance of pressure at 1 bar. The final production stage was carried out for 100 ns timescale with the LINCS algorithm [33] used for the generation of the structural conformations. The changes in the distance, H-bonds, RMSD, and Rg of both the complex systems were analyzed and the validation of the calculated binding energy was performed using the Molecular mechanics Poisson–Boltzmann surface area (MM-PBSA) protocols [34]. The RMSD and Rg were calculated on the basis of following formulas [18]:(1)$$RMSD\;\left(t_{1},\;t_{2}\right)={\left[\frac{1}{M}\overset{N}{\underset{i=1}{\sum }}m_{i{\left\|r_{i\;\left(t_{1}\right)}-r_{i\left(t_{2}\right)}\right\|}^{2}}\right]}^{\frac{1}{2}}$$
where $M=\overset{N}{\underset{i=1}{\sum }}m_{i}\;$and $r_{i\left(t\right)\;}$is the position of atom *i* at time *t*
(2)$$Rg={\left(\frac{\sum_{i}{\left\|r_{i}\right\|}^{2}m_{i}}{\sum_{i}m_{i}}\right)}^{\frac{1}{2}}$$
where *m_i_* is the mass of atom *i* and *r_i_* the position of atom *i* with respect to the center of mass of the molecule.

### 2.4. Bacterial Strains and Growth Conditions

Two *M. smegmatis* strains, differing in the levels of the *mmpS5-mmpL5* operon expression, were used in the study: the wild type (*w.t.*) *M. smegmatis mc2 155*, with a basal level of *mmpS5-mmpL5* expression, and *M. smegmatis atr9c*—a recombinant strain with a mutation in MSMEG_1380 gene (insC_8_), leading to over-expression of the *mmpS5-mmpL5* operon [14]. *M. smegmatis* were cultured in liquid Middlebrook 7H9 medium (Himedia, Mumbai, India) supplemented with oleic albumin dextrose catalase (OADC, Himedia, Mumbai, India), 0.1% Tween-80 (*v*/*v*), and 0.4% glycerol (*v*/*v*) in the Multitron incubator shaker (Infors HT, Basel, Switzerland) at 37 °C and 250 rpm, while soyabean-casein digest agar (M290, Himedia, Mumbai, India) was used as the solid media, with incubation at 37 °C.

### 2.5. Drug Susceptibility Testing by Paper-Disc Method

The paper-disc drug susceptibility assay was performed as described before [35]. Briefly, *M. smegmatis* cultures were grown overnight in Middlebrook 7H9 broth to mid-exponential phase (OD_600_ = 1.2), and afterwards were diluted in the proportion of 1:9:10 (culture:water:soyabean-casein digest agar) and 5 mL were poured as the top layer on Petri dishes with soyabean-casein digest agar. After the plates have dried for at least 30 min, paper discs with antimycobacterial agents (imidazo[1,2-b][1,2,4,5]tetrazine 3a [35], tryptanthrin (TRP) and 8-fluorotryptanthrin (PK31) [15]) subjected to MmpS5-MmpL5 efflux alone, paper discs with tested inhibitors alone, and paper discs with the combination of an antimycobacterial agent and an inhibitor were plated on the agar. Plates were incubated for 2 days at 37 °C, until the bacterial lawn was fully grown, and the growth inhibition halos were clearly visible. Growth inhibition halos were measured to the nearest 1 mm. The experiments were carried out as triplicates, the average diameter and standard deviation (SD) were calculated. Those differences that had no intersection of the SDs with the control were considered significant.

## 3. Results

### 3.1. Generation of MmpS5-MmpL5 Assemblies

The structures of both MmpS5 and MmpL5 were modeled using the previously available approaches described in the literature [16]. The model generated by Phyre2 using CusA (PDB ID-3K07 [36]) was selected for further analysis. The predicted model showed the confidence level of 100% and on the stereo-chemical validation using the Ramachandran plot showed 98.3% of the residues were occupied in the allowed regions. The 3-D model for MmpS5 with 100% confidence and identity of 58% as well as predicted with the complete chain of a template (PDB ID-2IW3 [37]) was considered. The respective model showed 98.5% of residues in the allowed regions of the Ramachandran plot. When the assessments of the non-bonded atomic interaction for the predicted models were performed using the ERRAT server [38], it was observed that the model of MmpS5 and MmpL5 showed the quality scores of 76.52 and 51.42, respectively, which were further improved through the structure optimizations. Furthermore, the complex of both MmpS5 and MmpL5 systems was obtained through protein-protein docking using ZDOCK. Around 2000 docked poses were generated and the residues for the interaction interface was selected by analyzing the corresponding sites in *E. coli* AcrB-AcrA protein complex [16]. The poses with the highest scores were selected for further analyses.

### 3.2. Selection of Highest Inhibitory Compounds

After successful generation of the 3-D model assembly, the 5968 anti-bacterial compounds were collected from the ASINEX repository. The structures of all the compounds were prepared for virtual screening using the LigPrep module. The substrate binding site was predicted using the COACH server which identified residues Leu442, Ile444, Glu445, Thr479, Asp511, Arg512, Ala513, Asp514, Asp515, Met516, Leu518, Gln519, Thr722, Glu724, Gly725, and Ile726 to be the part of the interaction pocket. The coordinates were fixed on these residues and all the inhibitor molecules were subjected to virtual screening using Autodock Vina. The 100 inhibitors with the highest free energies of binding were selected for further evaluations.

The QSAR modeling was further utilized for the filtration of the compounds with the highest inhibitory parameters. The inhibitors with anti-efflux activities against the protein of *M. smegmatis* were collected from the literature [25]. The collected 30 compounds are listed in Appendix A. The minimum inhibitory concentrations (MICs) were considered for this study and to obtain the uniformity, they were converted to corresponding pMIC using the following expression:(3)$$pMIC=-\log\left(\frac{MIC\left(\mu g/mL\right)}{Mw\left(gmol^{-1}\right)}\times 10^{-3}\right)$$
where the M_w_ is the Molecular weight.

The obtained set was then randomly converted into the training and the test sets. Thereafter in order to generate the QSAR model, the structures of all the collected inhibitors were drawn and then 3-D coordinates were generated. Then the “LigPrep” module was used which optimized the structure using the OPLS forcefield [39] and performed full energy minimization using the “Epik” approach [40]. After the generation of 3-D structures, the suitable descriptors were generated and QSAR modeling was performed using the “AutoQSAR” module present in the Schrodinger suite. The AutoQSAR uses the machine learning approaches and generated 10 models (Appendix A) with the highest R^2^ and Q^2^ values which are considered to be significant while evaluating the accuracy of the generated QSAR model. Model 6 was observed to be of the highest significance because of the observed closeness in the R^2^ and Q^2^ values of 0.873 and 0.843 respectively (Appendix A). The pMIC values predicted using model 6 was compared with the experimentally derived pMIC values and a relative closeness was observed indicating the higher efficiency of the developed model 6 (Appendix A).

On the basis of the outcomes generated from the virtual screening and QSAR studies, five compounds that showed the highest inhibitory effects were selected for further studies (Table 1). The BDD_27860195 and BDE_26593610 showed relatively higher inhibitory parameters. The BDD_27860195 was observed to interact with Gln190, Ser193, Lys436, Arg506, Lys507, Arg512, and Glu893 (Figure 1A) while BDE_26593610 showed Ser197, Lys436, Asn438, Lys507, Tyr508, Arg512, Met516, and Glu893 (Figure 1B). The Gln190 and Ser193 were observed to form the hydrogen bonding with the BDD_27860195. The respective inhibitor showed hydrophobic interactions with Phe255, Arg506, Lys507, and Tyr508, with the salt bridge formation occurred with Glu893. Whereas for BDE_26593610 it was noted that Lys507 and Glu893 were involved in the formation of hydrophobic, hydrogen bonding, and salt bridge.

### 3.3. Analyses of the Conformational Dynamics of the Assembled Systems

The generation of the whole MmpS5-MmpL5 assembly involved the modeling of MmpL5 into the trimeric conformation using the SymmDock server which included the docked inhibitor. Then the MmpS5 protein was mounted onto the trimer using the ZDOCK. After the successful creation of the assemblies, the residues involved in the membrane orientations were predicted using the PPM server (Figure 2).

Afterward the POPC membrane was modelled around the oriented residues and the systems were subjected to the 100 ns MD simulations in order to study the binding efficiencies of the selected inhibitors. The stability of the docked systems was evaluated in terms of projected RMSD values which indicated that the overall topology of the system has not changed because of the lower calculated values (Figure 3A). In comparison, it was observed that the MmpS5-MmpL5 docked system with BDD_27869195 achieved more stability than BDE_26593610 as lower RMSD values were calculated for the respective system. Both the systems showed the RMSD values fluctuating between 0.6–0.8 nm. Furthermore, the relative compactness of the systems was studied in light of plotted Rg values (Figure 3B). There is a significant difference between the Rg values obtained for both systems. The BDE_26593610 system obtained considerable higher compactness in the structural topology as compared to the BDD_27869195, with the calculated values observed around 3.2 nm after 10 ns. These observations indicated that the binding of the respective inhibitors impacted the structural topology of the MmpS5-MmpL5 system. Moreover, the nature of inhibitors binding was further analyzed using the pattern of calculated hydrogen bonding (H-bonds) as well as computed molecular distances between the protein and inhibitors. In the BDD_27869195 system, around five hydrogen bonds were observed in comparison to the BDE_26593610 system in which only three bonds were present. These observations were supported from the calculated distances with relatively higher closeness for BDD_27869195 system with values projected around 0.2 nm and little higher values computed for BDE_26593610 system.

The conformational stability of both the systems was compared using the Free Energy Landscapes (FEL). A considerable difference in FEL for both the systems was observed. A very narrow projection for the BDE_26593610 system was observed as compared to the BDD_27869195 system indicating the attainment of higher energetically favored conformation in the former system (Figure 4). Further analyses of binding affinities were carried out using the MMPBSA calculation between the protein and inhibitors (Table 2). For the BDD_27869195 system, the free energy of binding was calculated at around −317.108 kJ/mol which is significantly higher in comparison to the BDE_26593610 system in which the energy of −172.407 kJ/mol was observed. These observations indicated that both the studied compounds significantly inhibit the functionality of the MmpS5-MmpL5 system.

### 3.4. Exploring the Potential of BDD_27860195 and BDE_26593610 as Drugs Molecules

The drug likeliness of BDD_27860195 and BDE_26593610 was predicted by calculating the values of logP as well as by computing the probabilities of metabolism of these compounds on isoforms 3A4 and 3D6, as well as the cytotoxic effect to human cells using models available on the chemosophia [41]. The available models are based on the molecular interior based approach—3D QSAR CoMIn (Continual Molecular Interior analysis) algorithm [41]. It overlays molecules to maximize the coincidence of the potentials or the quantum functions at the junctions of the generalized lattice [42]. The potentials are the distribution of MERA atomic “matter” Equation (4), its derivative Equation (5) and their products with different weight factors ($w_{i}$):(4)$$\varphi_{j}=w_{ij}\alpha_{j}e^{-\beta_{j}r_{jm}^{2}}$$
(5)$$\varphi_{j}^{\prime }=-2w_{ij}\beta_{j}r_{jm}\alpha_{j}e^{-\beta_{j}r_{jm}^{2}}$$
where $w_{ij}$ is *i*-th weight factor of atom *j* (atomic charge, volume, number of occupied atomic orbits, number of unoccupied atomic orbits, HOMO and LUMO energies as well as the products of these weight factors), $r_{jm}$ is the distance of the atom *j* from the lattice junction *m*, and $\alpha_{j}$ and $\beta_{j}$ are explained in [43]. Afterwards, the relationships between bioactivity and descriptors based on linear reaction of neural network (LNN), or neural network with sigmoid neurons (NNSN) were established for the calculation of probability of the bioactivity expressed.

It was observed that the value of logP predicted using CoMIn models completely satisfies the Lipinski’s rule, namely logP < 5, being 1.67 for BDD_27860195 and 2.56 for BDE_26593610. The CoMIn prognosis showed good metabolic properties for both compounds, however, BDD_27860195 has more promising metabolic properties. On one hand, it showed the ability to interact with the presented targets before metabolism, while on another hand it is characterized by subsequent excretion from the body due to metabolism. The CoMIn prognosis shows moderate cytotoxicity of these compounds (Table 3).

### 3.5. BDE_26593610 and BDD_27860195 Can Inhibit M. smegmatis MmpS5-MmpL5 System In Vitro

We used a test-system of two *M. smegmatis* strains (the *w.t. M. smegmatis mc2 155* and the recombinant *M. smegmatis atR9c*) to assess the MmpS5-MmpL5 inhibitory potential of the selected compounds in vitro. To be classified as an active MmpL5 inhibitor, a compound should sensitize *M. smegmatis atr9c* strain (with overexpression of *mmpS5-mmpL5* operon) to anti-mycobacterial agents, subjected to MmpS5-MmpL5 efflux, when added together to a disc, producing a larger growth inhibition halo than the anti-mycobacterial agent alone, while a strong inhibitor might also sensitize the *mc2 155* strain (with only a basal level of *mmpS5-mmpL5* expression).

To exclude the possible toxic effect of the potential MmpS5-MmpL5 inhibitors, their sub-inhibitory concentrations were used in the study, thus the anti-mycobacterial activity of the compounds was assessed by the paper-disc method at 3 concentrations (10, 50 and 100 nmol/disc). All the tested inhibitors, except for LAS_52157603 were not toxic on *M. smegmatis* strains at concentrations up to 100 nmol/disc. LAS_52157603 produced a growth inhibition halo of 13 mm at 100 nmol/disc, and 9–10 mm at 50 nmol/disc. All the compounds were subsequently tested at concentrations of 10 and 50 nmol/disc in combination with anti-mycobacterial agents. For **3a** 400 nmol/disc were applied, while 30 nmol/disc were used for TRP and 10 nmol/disc for PK31.

The inhibitors have shown no effect at concentrations of 10 nmol/disc when combined with **3a** (Figure 5A). However, 2 compounds (BDE_26593610 and BDD_27860195) were active at the concentration of 50 nmol/disc, sensitizing *M. smegmatis atr9c* to **3a** (Figure 5B and Appendix A). LAS_52157603 has also sensitized *M. smegmatis atr9c* to **3a**, but it produced a growth-inhibition halo alone, thus this could be a synergistic cytotoxic effect, and this compound cannot be considered a specific MmpS5-MmpL5 inhibitor.

We additionally tested all the inhibitors at 50 nmol/disc (25 nmol/disc were used for LAS_52157603) for synergistic effect with tryptanthrins—another group of drugs affected by MmpS5-MmpL5 efflux. Compounds BDF_33196400, LAS_51205871 and LAS_52157603 showed no activity in MmpS5-MmpL5 inhibition in this experiment too, though LAS_52157603 was still able to produce a barely noticeable 6 mm growth inhibition halo alone, while BDE_26593610 and BDD_27860195 were active at the concentration of 50 nmol/disc, sensitizing *M. smegmatis atr9c* to TRP and its derivative PK31 (Figure 6 and Appendix A).

## 4. Discussion

The MmpS5-MmpL5 protein systems are significant for the drugs efflux mechanism utilized within the *Mycobacterium* species as well as in the intake of extracellular iron through the transport of siderophore molecules [16]. Due to the unavailability of the crystal structures, the molecular modeling techniques were explored for the development of 3-D models of MmpL5 proteins. The predicted structure contains 31 α-helices and 15 β-strands which are arranged in 3-D topology similar to the members belonging to the RND transporter protein family. The members of this family, functioning as efflux pumps, are involved in the development of multidrug resistance among different bacterial species [11]. After analyzing the structure of MmpL5 protein and comparing it with the member of the RND family, it was observed that the predicted model contains three structural domains which may be involved in the bridging of inner and outer bacterial membranes [11]. The components that hold the inner, as well as the outer membrane, are known as RND and outer membrane proteins respectively, while the periplasmic part forms the membrane fusion protein [11].

After the generation of the MmpL5 structure, its trimeric form was obtained using the SymmDock and the complete assemble of the MmpS5-MmpL5 system was created by docking the modeled MmpS5 system onto the trimeric MmpL5. Then the virtual screening with the library of 5968 anti-bacterial compounds was performed on the predicted ligand binding site on MmpL5 protein. The inhibitors with the highest free energy of binding were ranked. To achieve the higher accuracy in the process of selecting the most suitable inhibitors, QSAR models were generated using the AutoQSAR modules of Schrodinger. The combination of both free energy of binding and the predicted pMIC values using the QSAR modeling was used for the selected molecules with highest inhibitory efficiency (listed in Table 1). The inhibitors were observed to be binding into the interaction site of the MmpL5 protein favorably. To validate the docking parameters, the 100 ns MD simulations were performed on the selected BDD_27860195 and BDE_26593610 based MmpS5-MmpL5 systems. It was observed that the selected inhibitors bounded to the MmpS5-MmpL5 system with significant interaction energy parameters.

Moreover, we were able to show that BDE_26593610 and BDD_27860195 can be considered active MmpS5-MmpL5 inhibitors, as they were able to sensitize in vitro *mmpS5-mmpL5* overexpressing *M. smegmatis atr9c* strain to the tested imidazo[1,2-b][1,2,4,5]tetrazine and the tryptanthrins, while *M. smegmatis mc2 155* (with a *w.t.* expression level of the *mmpS5-mmpL5* operon) was also sensitized by these compounds to PK31, confirming that the MmpS5-MmpL5 efflux specifically provides a basal level of resistance to this tryptanthrin derivative [15].

## 5. Conclusions

The advent of the COVID-19 pandemic, which in combination with the existing drug resistance conditions in TB patients, is limiting the current regime of treatment in the scarcely available medical infrastructure. The active efflux systems in the *Mycobacterium* species are among the major causes leading to the development of multi-drug resistance conditions and their low concentration activities are associated with the generation of drug tolerant mutations. Therefore, a shift is needed for the development of inhibitors targeting bacteria efflux systems. As a result, the MmpS5-MmpL5 system which is an efflux system present in *Mycobacterium* species was selected as a potential target for the identification of suitable inhibitors. The virtual screening approach in combination with the QSAR modeling enables the identification of anti-bacterial compounds with the highest inhibitory values. The BDD_27860195 and BDE_26593610 showed most significant binding to the MmpS5-MmpL5 systems, which were validated using 100 ns MD simulations. The probability of using BDD_27860195 and BDE_26593610 as potential drugs were also explored, which showed suitable metabolic properties, restrained cytotoxicity towards human cells, and logP value satisfying Lipinski’s rule. Furthermore, the in vitro inhibitory studies showed that the respective molecules may function as active inhibitors of *M. smegmatis* MmpS5-MmpL5-mediated efflux. The outcome of this study can further be used in the formulation of potent inhibitory molecules and will contribute significantly to the process of drug design and discovery of anti-TB compounds.

## Figures and Tables

**Figure 1 biomedicines-10-00333-f001:**
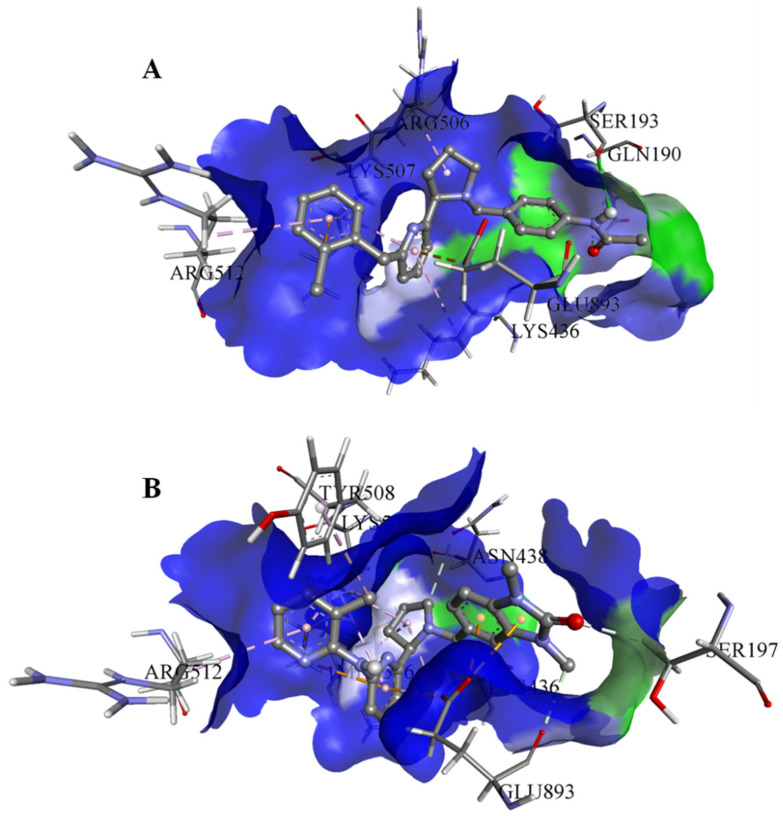
The graphical representation of the MmpS5-MmpL5 docked complexes with (**A**) BDD_27860195 and (**B**) BDE_26593610.

**Figure 2 biomedicines-10-00333-f002:**
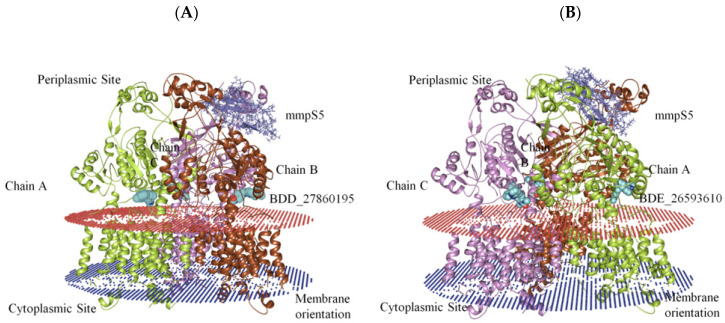
The graphical representation of assembled mmpS5-mmpL5 with compounds BDD_27869195 and BDE_26593610. (**A**) shows MmpS5-MmpL5 with compound BDD_27869195 (Blue). (**B**) shows MmpS5-MmpL5 with compound BDE_26593610 (Blue).

**Figure 3 biomedicines-10-00333-f003:**
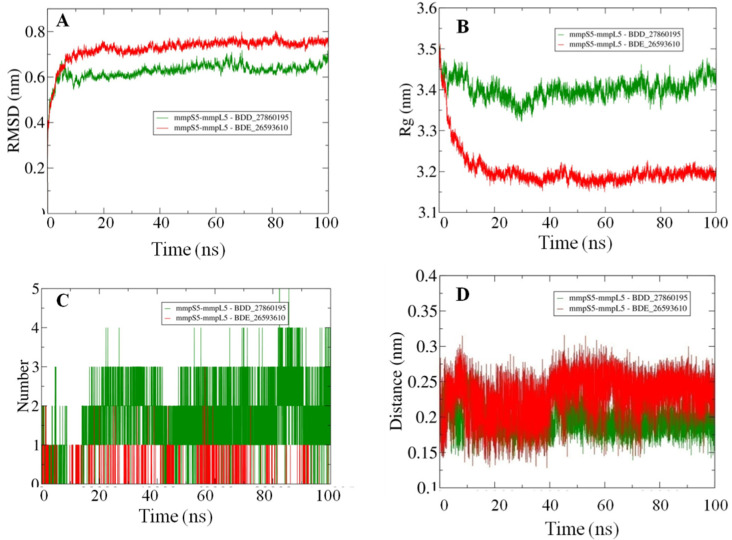
The outcomes of 100 ns MD simulations showing (**A**) Changes in the RMSD values reflecting the variations in the topologies for both the systems (Standard Deviation (SD), BDD_27869195 = 0.035 nm, BDE_26593610 = 0.05 nm). (**B**) Fluctuation in the Rg values shows the changes in the compactness of the studied systems (SD, BDD_27869195 = 0.029 nm, BDE_26593610 = 0.07 nm). (**C**) Dynamics of the H-bond patterns were observed for both systems (SD, BDD_27869195 = 0.89, BDE_26593610 = 0.29). (**D**) Differences in the calculated distance values for both the studied systems (SD, BDD_27869195 = 0.02 nm, BDE_26593610 = 0.03 nm).

**Figure 4 biomedicines-10-00333-f004:**
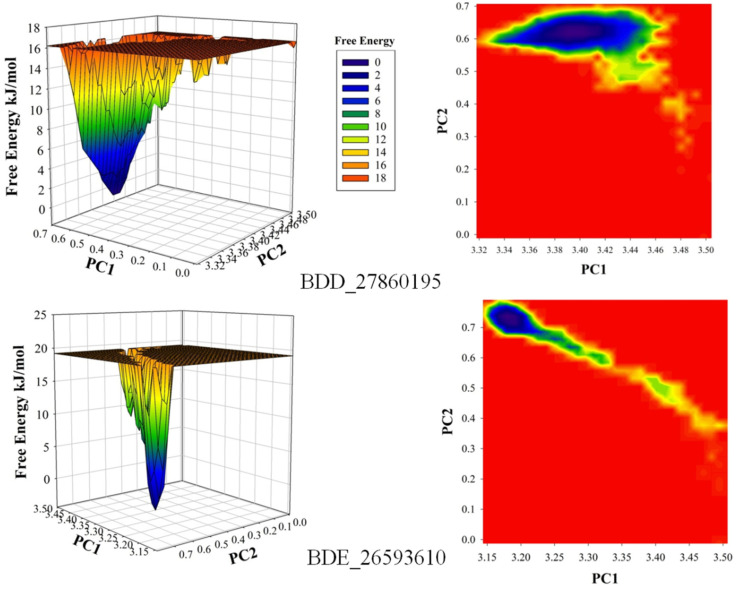
The calculated free energy landscapes showing the changes in the conformational stability (PC1 is RMSD and PC2 is Rg).

**Figure 5 biomedicines-10-00333-f005:**
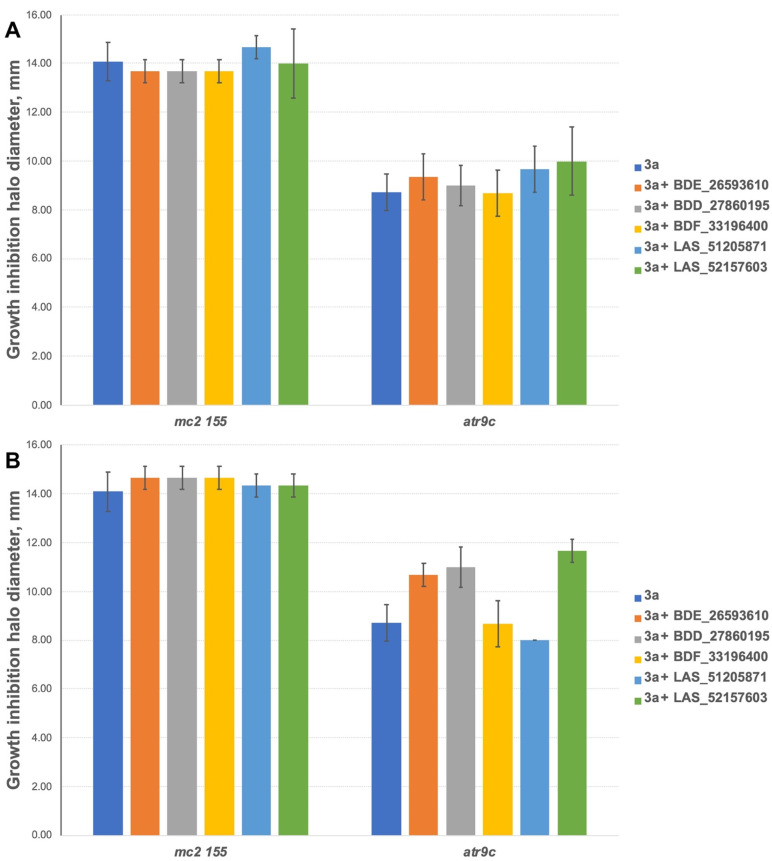
Growth inhibition halos, produced by **3a** and its combination with MmpS5-MmpL5inhibitors: (**A**) at 10 nmol/disc, and (**B**) at 50 nmol/disc. The error bars represent the SDs.

**Figure 6 biomedicines-10-00333-f006:**
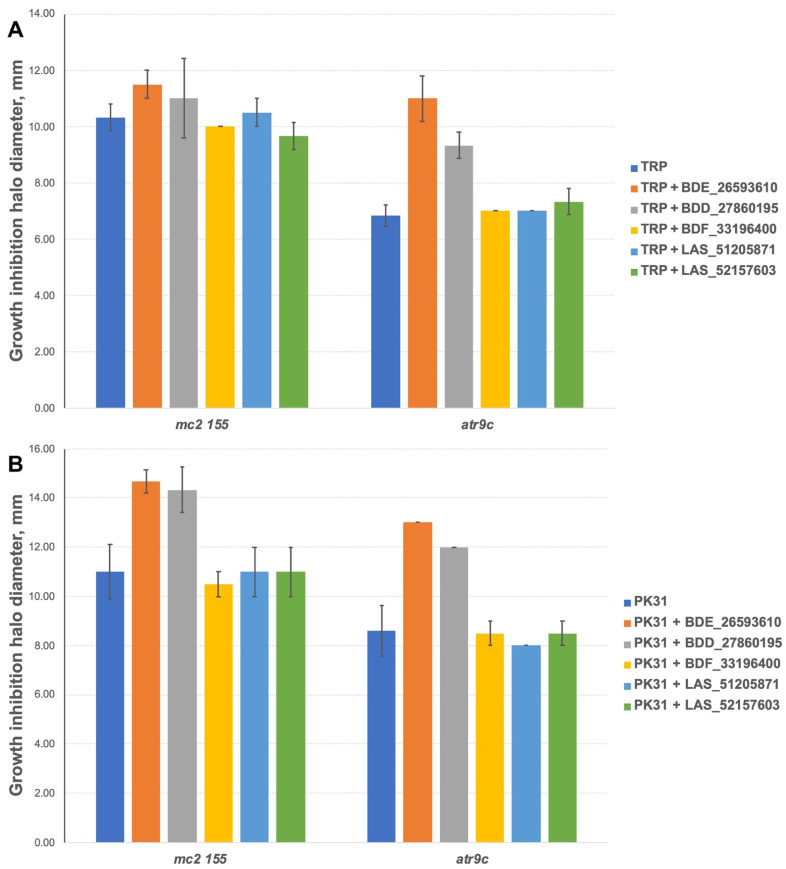
Growth inhibition halos, produced by the combination of 50 nmol/disc MmpS5-MmpL5inhibitors (25 nmol/disc for LAS_52157603) with (**A**) TRP and (**B**) PK31. The error bars represent the SDs.

**Table 1 biomedicines-10-00333-t001:** List of selected inhibitors from the set of 5968 ASINEX antibacterial compounds on the basis of the outcomes generated from the virtual screening and QSAR studies.

S. No	ASINEX ID	Free Energy of Binding (Kcal/mol)	Predicted pMIC Values from QSAR	Structure
1.	BDD_27860195	−9.5	4.772	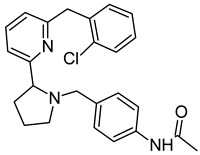
2.	BDE_26593610	−9.5	4.863	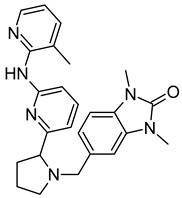
3.	BDF_33196400	−9.5	4.368	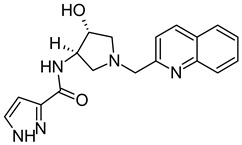
4.	LAS_51205871	−9.6	4.208	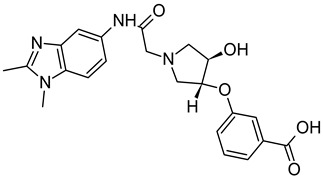
5.	LAS_52157603	−9.5	4.526	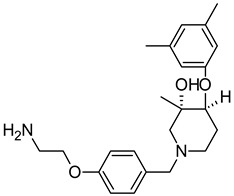

**Table 2 biomedicines-10-00333-t002:** List of MMPBSA based free energy parameters for both MmpS5-MmpL5 inhibitor systems.

S. No.	System	MMPBSA Based Energies (kJ/mol)				
		ΔE (vdW)	ΔE (Elec)	ΔG (Polar)	ΔG (Non-Polar)	ΔG (Binding)
1.	BDD_27860195	−307.339	−8.355	20.617	−22.031	−317.108
2.	BDE_26593610	−185.701	−2.025	31.273	−15.954	−172.407

**Table 3 biomedicines-10-00333-t003:** CoMIn predicted properties for BDD_27860195 and BDE_26593610.

S. No.	System	CoMIn Predicted Properties			
		logP	P(2D6)	P(3A4)	P(CYT)
1.	BDD_27860195	1.67	0.505	0.633	0.232
2.	BDE_26593610	2.56	0.701	0.740	0.208

## Data Availability

The data presented in this study are available in www.mdpi.com/article/10.3390/biomedicines10020333/s1.

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
