# Peer review of "Repurposing Based Identification of Novel Inhibitors against MmpS5-MmpL5 Efflux Pump of Mycobacterium smegmatis: A Combined In Silico and In Vitro Study"

_biomedicines, 2022, doi:10.3390/biomedicines10020333_

Round 1

Reviewer 1 Report

Dear authors,

Here are my comments and suggestions on what can be improved to enrich it:

*ABSTRACT*

Lines 34-35: A strong statement based on theoretical and in vitro analysis using a model for a different bacteria. I suggest changing the very emphatic tone.  

*INTRODUCTION*

Very nice introduction.

Lines 94-95 have a similar statement as observed in lines 34-35. Same suggestion.

*2. Materials and Methods*

2.1.

Line 101: reference for the Phyre2 server is missing.

Line 105: which steric clashes were removed? Couldn’t these clashes be avoided in the homology construction?

2.2.

Line 116: reference and date of access are missing for ASINEX screening libraries. Please add to the bibliography.

Line 117: missing reference for Schrodinger 2020.2 suite.

Line 118 (and line 196): AutoDock Vina’s name typos.

Also, about the AutoDock Vina: what were the parameters used? Score functions? Did you use flexible side chains? What was the partial charges model applied? Please, explicit the docking details like was done in the MD part.

Line 123: which specific machine learning algorithms were utilized? What are its advantages and disadvantages?

2.3.

Why do the authors use pure POPC as a membrane model for bacteria? What are the advantages and weaknesses of the model? And how the coating on the cell surface of the Mycobacterium tuberculosis can affect the readings of the simulation?

Line 132: “the negative charges in the systems.” The POPC lipid is zwitterionic. Is it correct to assume that the negative charge of the systems is from the proteins? Please add the net charge of the models.

Also, why was K+ the choice as counter ions? Why not Na+?

Lines 138-141: There are missing references for Berendsen and Nose-Hoover thermostats, Parrinello-Rahman barostat, and the LINCS algorithm. More detail of each coupling should be added. Which time-step was used? What are the system sizes?

Line 142: RMSD and Rg were not defined.

*RESULTS*

References are missing for the PDBs in lines 174 and 178.

Line 182: “quality scores of 76.52 and 51.42 respectively, which were further improved through the structure optimizations”. Did the authors measure the quality scores again after the optimizations? If yes, what were the new values?

Line 201: Table S1 mentioned in the text is not presented in the Supplemental Material or the text. The SM has only two Figures (photos). Not sure if it was not uploaded or it was an issue with the file that was available to download. The same happens with Table S2 and Table S3.

Line 204: Please point to Equation 1. And define its terms. What is Mw? Molecular weight?

Line 207-211: There are missing references for the OPLS force field and Epik approach.

Line 211: what is the “machine learning approach”? What type of machine was used? Which algorithm?

Line 214: Model 6 is not presented. Was it in the Supplemental Material list?

Table 1: units are missing. Please, check the structures and the space. Structures 1, 4, and 5 have issues of superposition. Impossible to correctly visualize the systems.

What means BDD, BDE, BDF, and LAS?

Lines 224-228: please refer to each part of the Figure (A and B) for better identification. Also, for Figure 1, I recommend adopting other representations or color schemes to visualize better. Gray over gray is not so easy to see.

Figure 2 has a typo in “compound” – line 244.

The authors perform 100 ns simulations. But were the membranes equilibrated? If yes, what are the parameters that characterize that? Area per lipid headgroup, membrane thickness, etc.?

Also, are 100 ns simulations enough? These parameters could show that. For example, the RMSD for one protein seems to be increasing (Figure 3 A).

What are the errors associated with the calculations and measurements? Please add that to text and to Figure 3 where it fits.

In Figure 3C, I suggest changing the representation to bars or symbols.

Line 273: how the FEL were generated? This also implies explaining what are PC1 and PC2 in Figure 4. No explanation or definition was provided.

Line 281: energy units missing.

3.4.

Line 293: reference for the 3D QSAR CoMIn algorithm is missing.

Equations 4 and 5 (should be 2 and 3?) just showed up, and some of the contexts here could be added in the Methods part. The authors also mentioned that the parameters are explained in another paper. For an easier understanding of the readers, I suggest adding a short definition here.

*SUPPLEMENTARY MATERIALS*

Missing Tables S1, S2, and S3.

Author Response

Reviewer 1

*ABSTRACT*

Lines 34-35: A strong statement based on theoretical and in vitro analysis using a model for a different bacteria. I suggest changing the very emphatic tone.  

 Response:  Revised as per the reviewer suggestions.

*INTRODUCTION*

Very nice introduction.

Lines 94-95 have a similar statement as observed in lines 34-35. Same suggestion.

 Response:  Revised as per the reviewer suggestions.

*2. Materials and Methods*

2.1.

Line 101: reference for the Phyre2 server is missing.

Response: The reference for Phyre 2 is already included as reference number 17.

Line 105: which steric clashes were removed? Couldn’t these clashes be avoided in the homology construction?

Response: Steric clashes arise due to the unnatural overlap if two nonbonding atoms were removed. It is not possible to avoid these clashes in homology construction as quite low identity was observed between the target and template. That's why standard MD simulation procedure was used to remove these steric clashes.

2.2.

Line 116: reference and date of access are missing for ASINEX screening libraries. Please add to the bibliography.

Response: Reference has been added in the revised manuscript as per reviewer suggestions.

Line 117: missing reference for Schrodinger 2020.2 suite.

Response: The reference is added in the revised manuscript.

Line 118 (and line 196): AutoDock Vina’s name typos.

Response: The typos are corrected as per the reviewer suggestions.

Also, about the AutoDock Vina: what were the parameters used? Score functions? Did you use flexible side chains? What was the partial charges model applied? Please, explicit the docking details like was done in the MD part.

Response: The details are added in the revised manuscript.

Line 123: which specific machine learning algorithms were utilized? What are its advantages and disadvantages?

Response: The AutoQSAR uses machine learning algorithms such as Multiple Linear Regression, Partial least squares regression, Principal components regression, Kernel-based PLS (KPLS), Naive Bayes classification and Ensemble recursive partitioning. For our study, KPLS methods showed most promising results. Kernel partial least squares (KPLS) method is among the popular techniques for regression of complex non-linear data sets, with the modeling is performed by mapping the data in a higher dimensional feature space through the kernel transformation. The disadvantage of such a transformation is, however, that information about the contribution of the original variables in the regression is lost.

2.3.

Why do the authors use pure POPC as a membrane model for bacteria? What are the advantages and weaknesses of the model? And how the coating on the cell surface of the Mycobacterium tuberculosis can affect the readings of the simulation?

Response: On the basis of the information available in the Literature (Reference 16) regarding the modeling of MmpS5-MmpL5 efflux pump, the POPC was selected for the current study. The POPC was used for mimicking the in vivo environmental conditions of the protein (Reference 16). The aim of the study was to do experimental and computational both on the M. tuberculosis systems, but due to experimental shortcoming we performed it on the computational system, thats why POPC was used. The other objective is that this study can be used as input in other studies aimed at designing novel inhibitors against the M. tuberculosis.    

Line 132: “the negative charges in the systems.” The POPC lipid is zwitterionic. Is it correct to assume that the negative charge of the systems is from the proteins? Please add the net charge of the models.

Response: Yes, the charges are coming from the protein molecules as POPC lipid is zwitterionic. The net charge of all the system is zero as it is neutralized by adding the counter ions.

Also, why was K+ the choice as counter ions? Why not Na+?

Response: We have used the defaults of the CHARMMGUI for the system preparation. Their first choice is K+ ions. Another reason we found in the literature is that the smaller ions such as Na+ do not bind to these K+ sites in a thermodynamically favorable way

Lines 138-141: There are missing references for Berendsen and Nose-Hoover thermostats, Parrinello-Rahman barostat, and the LINCS algorithm. More detail of each coupling should be added. Which time-step was used? What are the system sizes?

Response: The references are added as per the suggestion of the reviewer.

Line 142: RMSD and Rg were not defined.

Response: The expression for calculating the RMSD and Rg are defined in the revised manuscript.

*RESULTS*

References are missing for the PDBs in lines 174 and 178.

Response: The references are added in the revised manuscript.

Line 182: “quality scores of 76.52 and 51.42 respectively, which were further improved through the structure optimizations”. Did the authors measure the quality scores again after the optimizations? If yes, what were the new values?

Response: Yes, the quality scores were calculated and was observed to be around 78.4519 and 69.5829 respectively for MmpS5 and MmpL5.

Line 201: Table S1 mentioned in the text is not presented in the Supplemental Material or the text. The SM has only two Figures (photos). Not sure if it was not uploaded or it was an issue with the file that was available to download. The same happens with Table S2 and Table S3.

Response: All the Supplementary information is included in the revisions.

Line 204: Please point to Equation 1. And define its terms. What is Mw? Molecular weight?

Response: The terms are defined in the revised manuscript.

Line 207-211: There are missing references for the OPLS force field and Epik approach.

Response: Respective references are added in the revised manuscript.

Line 211: what is the “machine learning approach”? What type of machine was used? Which algorithm?

Response: The AutoQSAR uses machine learning algorithms such as Multiple Linear Regression, Partial least squares regression, Principal components regression, Kernel-based PLS (KPLS), Naive Bayes classification and Ensemble recursive partitioning. For our study, KPLS methods showed most promising results. Kernel partial least squares (KPLS) method is among the popular techniques for regression of complex non-linear data sets, with the modeling is performed by mapping the data in a higher dimensional feature space through the kernel transformation. The disadvantage of such a transformation is, however, that information about the contribution of the original variables in the regression is lost.

Line 214: Model 6 is not presented. Was it in the Supplemental Material list?

Response: Yes it is included in the Table S2 of the supplementary materials. It is now provided in the revisions.

Table 1: units are missing. Please, check the structures and the space. Structures 1, 4, and 5 have issues of superposition. Impossible to correctly visualize the systems.

Response: The respective units are added in the revised manuscript.

What means BDD, BDE, BDF, and LAS?

Response: These are the notations used in the ASINEX dataset. There are no description provided regarding the annotations on ASINEX website.

Lines 224-228: please refer to each part of the Figure (A and B) for better identification. Also, for Figure 1, I recommend adopting other representations or color schemes to visualize better. Gray over gray is not so easy to see.

Response: The Figure 1 is revised as per suggestion. The Figure notation in the text is also revised.

Figure 2 has a typo in “compound” – line 244.

Response: The typo is corrected.

The authors perform 100 ns simulations. But were the membranes equilibrated? If yes, what are the parameters that characterize that? Area per lipid headgroup, membrane thickness, etc.?

Response: Yes membranes were equilibrated in six steps as per the inputs are generated using the CARMMGUI server. We used the Area per lipid headgroup analysis in order to observe the changes.

Also, are 100 ns simulations enough? These parameters could show that. For example, the RMSD for one protein seems to be increasing (Figure 3 A).

Response: We agree with the reviewer that more time should be included but each system is generating around 1TB of data. We have limited space in the server.

What are the errors associated with the calculations and measurements? Please add that to text and to Figure 3 where it fits.

Response: The Standard Deviation (DS) is calculated for all the plotted parameters in Figure 3. The values are added in the revised manuscript. The lower values of the SD indicates that the attainment of lower calculation errors.   

In Figure 3C, I suggest changing the representation to bars or symbols.

Response: We are comparing two systems here which are clearly visible, therefore there is no need to change the representation. If we have higher number of systems then we agree with the reviewer to provide the changes in the H-bond pattern in the form of bar or symbols.

Line 273: how the FEL were generated? This also implies explaining what are PC1 and PC2 in Figure 4. No explanation or definition was provided.

Response: The FEL was generated using the " gmx sham" in combination with publically available PERL and PYTHON scripts. The PC1 is the RMSD and PC2 is Rg. The description is added in Figure 4)

Line 281: energy units missing.

Response: The unit is added in the revised manuscript

3.4.

Line 293: reference for the 3D QSAR CoMIn algorithm is missing.

Response: Suitable reference is added in the revised manuscript.

Equations 4 and 5 (should be 2 and 3?) just showed up, and some of the contexts here could be added in the Methods part. The authors also mentioned that the parameters are explained in another paper. For an easier understanding of the readers, I suggest adding a short definition here.

Response: We are thankful for the reviewer for providing the insights. But, the equations are placed after thorough discussion. The unnecessary description of the terminologies will confuse the reader as our manuscript is not aimed to go deeper in that direction.

Reviewer 2 Report

The manuscript entitled “Repurposing based identification of novel inhibitors against MmpS5-MmpL5 efflux pump of Mycobacterium smegmatis: A combined in silico and in vitro study” by Mohd Shahbaaz et al. demonstrates an interesting approach to search for new compounds useful to improve antibiotics efficacy against resistant M. tuberculosis via inhibiting MmpS5-MmpL5 efflux pump. Although I very doubt if it is possible to predict structures of active EPIs for transport proteins such complicated as RND-containing complexes using structure-based design, I appreciate the studies presented because very few lines of evidence are dedicated to this pump that undoubtedly is an important protein target in the fight against drug-resistant bacteria. Furthermore, in contrast to the widely elaborated AcrAB-TolC  pump model in E. coli, it is challenging to model this transporter (MmpS5-MmpL5) in the absence of the crystallographic structure in PDB. For these reasons, I recommend this manuscript for publishing in Biomedicines journal, but prior the following point should be improved and addressed:

  1. Table 1. What unit of MIC was used to calculate pMIC values presented? M, mM or mg/mL, nmol/disc or other? It should be mentioned at the table.

  1. Table 1. The structure of compounds in the table are untidy drawn, especially, BDD 27860195 and LAS 51205871. They must be improved according to 2D-chemical structure standards (e.g. ACS document style)

  1. Lines 324-335. Authors used the highest concentration 50 nmol/disc of inhibitors in the combination with antibiotics, while the inhibitors were not toxic for these bacterial strains up to 100 nmol/disc. The concentration 50 nmol/disc is a risky one. It is advised to use 25 nmol/disc as the highest safe concentration of tested inhibitors. Thus, the inhibitory activity of compounds BDE 26593610 and BDD 27860195 at the concentration of 50  nmol/disc is rather doubtful.

  1. Lines 336-337, It is described: ” LAS 52157603 has also sensitized smegmatis atr9c to 3a, but it produced a growth-inhibition halo alone”. What was the concentration of LAS 52157603 used (in combination with the antibiotic) that demonstrated bacteria growth inhibiting properties? If it was lower than 12 nmol/disc, the compound can be considered as real inhibitory properties with higher probability than that for BDE 26593610 and BDD 27860195.

Overall, I advise Authors to use microdillution assays rather than disc methods for this kind of studies in the future.

Author Response

Reviewer 2

  1. Table 1. What unit of MIC was used to calculate pMIC values presented? M, mM or mg/mL, nmol/disc or other? It should be mentioned at the table.

Response: The mg/mL was used for the calculation of pMIC

  1. Table 1. The structure of compounds in the table are untidy drawn, especially, BDD 27860195 and LAS 51205871. They must be improved according to 2D-chemical structure standards (e.g. ACS document style)

Response: The presentation is changed to ACS style

  1. Lines 324-335. Authors used the highest concentration 50 nmol/disc of inhibitors in the combination with antibiotics, while the inhibitors were not toxic for these bacterial strains up to 100 nmol/disc. The concentration 50 nmol/disc is a risky one. It is advised to use 25 nmol/disc as the highest safe concentration of tested inhibitors. Thus, the inhibitory activity of compounds BDE 26593610 and BDD 27860195 at the concentration of 50  nmol/disc is rather doubtful.
  2. Lines 336-337, It is described: ” LAS 52157603 has also sensitized smegmatis atr9c to 3a, but it produced a growth-inhibition halo alone”. What was the concentration of LAS 52157603 used (in combination with the antibiotic) that demonstrated bacteria growth inhibiting properties? If it was lower than 12 nmol/disc, the compound can be considered as real inhibitory properties with higher probability than that for BDE 26593610 and BDD 27860195.

Response: All the compounds were initially tested at 10, 50 and 100 nmol/disc, and all except for LAS 52157603 produced no growth inhibition halos at 100 nmol/disc, showing that it was still a subinhibitory concentration, thus we considered 50 nmol/disc a safe concentration for most of the compounds. We still tested 2 concentrations of the inhibitors: 10 and 50 nmol/disc. However, the LAS 52157603 still produced a growth inhibition halo (about 9 mm at 50 nmol/disc), and had rather a synergistic cytotoxic effect.

It was not mentioned in the text, but we did use 25 nmol/disc for LAS 52157603 in our tests with tryptanthrins. We double checked our raw data and now added it to the text on lines 358-362 and in the caption for Figure 6. LAS 52157603 was not able to produce a larger growth inhibition halo in combination with 3a at 10 nmol/disc (subinhibitory concentration), and with tryptanthrins at 25 nmol/disc, while a barely visible 6 mm growth inhibition halo could still be observed for this compound alone, thus it can’t be considered more active than BDE 26593610 and BDD 27860195 in terms of MmpS5-MmpL5 inhibition.

Reviewer 3 Report

In the presented manuscript, Shahbaaz and colleagues used an in silico approach to identify inhibitors of the MmpS5-MmpL5 efflux pump involved in Mycobacterium smegmatis drug resistance. Two compounds showed the characteristics of potential drugs and sensitized M. smegmatis  to selected antimicrobial agents in vitro. I only have minor comments.

Lines 161-162 – please check whether this sentence is correct.

Line 324 - Wouldn't it be better to write "To exclude the possible toxic effect of inhibitors” instead of "To exclude the possible synergistic…"?

Line 384 – for clarity, the full name of the strain should be used -  M. smegmatis mc2 155.

Supplementary materials contain only Figures S1 and S2, and Tables S1-S3 are missing.

Author Response

Reviewer 3

  1. Lines 161-162 – please check whether this sentence is correct.

Response: The sentence has been revised to be clearer.

  1. Line 324 - Wouldn't it be better to write "To exclude the possible toxic effect of inhibitors” instead of "To exclude the possible synergistic…"?

Response: The sentence has been revised as suggested.

  1. Line 384 – for clarity, the full name of the strain should be used -  M. smegmatis mc2 155.

Response: The full name of the strain is now used as suggested.

  1. Supplementary materials contain only Figures S1 and S2, and Tables S1-S3 are missing.

Response: All the Supplementary information is included in the revisions.

Round 2

Reviewer 1 Report

Dear authors,

The revisions are great and I would like to commend the work.
My minor suggestion would be to include the KPLS method and its description (that you kindly sent in the answer) in the Methodology part. This would make that part more complete.

"For our study, KPLS methods showed most promising results. Kernel partial least squares (KPLS) method is among the popular techniques for regression of complex non-linear data sets, with the modeling is performed by mapping the data in a higher dimensional feature space through the kernel transformation."

Best regards.

Author Response

We updated the manuscript at line 279 (methods section) with the following text:

"For our study, the Kernel partial least squares (KPLSmethods showed most promising results. The KPLS method is among the popular techniques for regression of complex non-linear data sets, with the modeling is performed by mapping the data in a higher dimensional feature space through the kernel transformation. The disadvantage of such a transformation is, however, that information about the contribution of the original variables in the regression is lost."